# Are Spiking Neural Networks more expressive than Artificial Neural Networks?

## Abstract

This article studies the expressive power of spiking neural networks with firing-time-based information encoding, highlighting their potential for future energy-efficient AI applications when deployed on neuromorphic hardware. The computational power of a network of spiking neurons has already been studied via their capability of approximating any continuous function. By using the Spike Response Model as a mathematical model of a spiking neuron and assuming a linear response function, we delve deeper into this analysis and prove that spiking neural networks generate continuous piecewise linear mappings. We also show that they can emulate any multi-layer (ReLU) neural network with similar complexity. Furthermore, we prove that the maximum number of linear regions generated by a spiking neuron scales exponentially with respect to the input dimension, a characteristic that distinguishes it significantly from an artificial (ReLU) neuron. Our results further extend the understanding of the approximation properties of spiking neural networks and open up new avenues where spiking neural networks can be deployed instead of artificial neural networks without any performance loss.

## 1 Introduction

Despite the remarkable success of deep neural networks (ANNs) [12], the downside of training and inferring on large deep neural networks implemented on classical digital hardware lies in their substantial time and energy consumption [23]. The rapid advancement in the field of neuromorphic computing allows for both analog and digital computation, energy-efficient computational operations, and faster inference ([21], [2]). In practice, a neuromorphic computer is typically programmed by deploying a network of spiking neurons (SNNs) [21], i.e., programs are defined by the structure and parameters of the neural network rather than explicit instructions.

SNNs are more biologically realistic as compared to ANNs, as they involve neurons transmitting information asynchronously through spikes to other neurons [9]. Different encoding schemes enable spiking neurons to represent analog-valued inputs, broadly categorized into rate coding (spike count) and temporal coding (spike time) ([8], [17]). In this work, we assume that information is encoded in the precise timing of a spike. The event-driven nature and the sparse information propagation through relatively few spikes enhance system efficiency by lowering computational demands and improving energy efficiency.

It is intuitively clear that the described differences in the processing of the information between ANNs and SNNs should also lead to differences in the computations performed by these models. Several groups have analyzed the expressive power of ANNs from the perspective of approximation theory ([24], [4], [11], [20]) and by quantifying the number of the linear regions ([10], [18]). At the same time, few attempts have been made that aim to understand the computational power of SNNs. ([13], [3]) showed that continuous functions can be approximated to arbitrary precision using

Submitted to Unifying Representations in Neural Models Workshop (UniReps 2023). Do not distribute.

SNNs in temporal coding. It has also been shown that spiking neurons can emulate Turing machines, arbitrary threshold circuits, and sigmoidal neurons ([15], [16]).

In the simplest of settings considered in [14], there remains a lack of a comprehensive theory that completely quantifies the approximation capabilities of SNNs. In an attempt to follow up along the lines of previous works ([14], [18], [22], [19]), we aim to extend the theoretical understanding that characterizes the differences and similarities in the expressive power between a network of spiking and artificial neurons employing a piecewise-linear activation function. Specifically, we aim to determine if SNNs possess the same level of expressiveness as ANNs in their ability to approximate various function spaces and in terms of the number of linear regions they can generate. The main results in Section 3 are centered around the comparison of expressive power between SNNs and ANNs.

## 2  Spiking neural networks

In neuroscience literature, several mathematical models exist that describe the generation and propagation of action-potentials. To study the expressivity of SNNs, the main principles of a spiking neuron are condensed into a (simplified) mathematical model, where certain details about the biophysics of a biological neuron are neglected. In this work, to analyze SNNs, we employ the noise-free version of the Spike Response Model (SRM) [7]. We assume a linear response function, where additionally each neuron spikes at most once to encode information through precise spike timing. This in turn simplifies the model and also makes the mathematical analysis more feasible for larger networks as compared to other models where spike dynamics are described by differential equations.

**Definition 1.** *A spiking neural network $\Phi$ is a (simple) finite directed graph $(V, E)$ and consists of a finite set $V$ of spiking neurons, a subset $V_{in} \subset V$ of input neurons, a subset $V_{out} \subset V$ of output neurons, and a set $E \subset V \times V$ of synapses. Each synapse $(u, v) \in E$ is associated with a synaptic weight $w_{uv} \geq 0$, a synaptic delay $d_{uv} \geq 0$, and a response function $\varepsilon_{uv} : \mathbb{R}^+ \to \mathbb{R}$. Each neuron $v \in V \setminus V_{in}$ is associated with a firing threshold $\theta_v > 0$, and a membrane potential $P_v : \mathbb{R} \to \mathbb{R}$,*

$$P_v(t) := \sum_{(u,v) \in E} \sum_{t_u^f \in F_u} w_{uv} \varepsilon_{uv}(t - t_u^f), \tag{1}$$

*where $F_u := \{t_u^f : 1 \leq f \leq n \text{ for some } n \in \mathbb{N}\}$ denotes the set of firing times of a neuron $u$, i.e., times $t$ whenever $P_u(t)$ reaches $\theta_u$ from below.*

In general, the membrane potential also includes the *threshold function* $\Theta_v : \mathbb{R}^+ \to \mathbb{R}^+$, that models the refractoriness effect. However, we assume that each neuron fires at most once, i.e., information is encoded in the firing time of single spikes. Thus, in Definition 1, the refractoriness effect can be ignored and the contribution of $\Theta_v$ is modelled by the constant $\theta_v$. Moreover, the single spike condition simplifies (1) to

$$P_v(t) = \sum_{(u,v) \in E} w_{uv} \varepsilon_{uv}(t - t_u), \quad \text{where } t_u = \inf_{t \geq \min_{(z,u) \in E} \{t_z + d_{zu}\}} P_u(t) \geq \theta_u. \tag{2}$$

The *response function* $\varepsilon_{uv}$ models the impact of a spike from a presynaptic neuron $u$ on the membrane potential of a postsynaptic neuron $v$ [7]. A biologically realistic approximation of $\varepsilon_{uv}$ is a delayed $\alpha$ function [7], which is non-linear and leads to intractable problems when analyzing the propagation of spikes through an SNN. Hence, following [15], we consider a simplified response and only require $\varepsilon_{uv}$ to satisfy the following condition:

$$\varepsilon_{uv}(t) = \begin{cases} 0, & \text{if } t \notin [d_{uv}, d_{uv} + \delta], \\ s \cdot (t - d_{uv}), & \text{if } t \in [d_{uv}, d_{uv} + \delta], \end{cases} \quad \text{where } s \in \{+1, -1\} \text{ and } \delta > 0. \tag{3}$$

The parameter $\delta$ is some constant assumed to be the length of a linear segment of the response function. The variable $s$ reflects the fact that biological synapses are either *excitatory* or *inhibitory* and the *synaptic delay* $d_{uv}$ is the time required for a spike to travel from $u$ to $v$. Inserting condition (3) in (2) and setting $w_{uv} := s \cdot w_{uv}$, i.e., allowing $w_{uv}$ to take arbitrary values in $\mathbb{R}$, yields

$$P_v(t) = \sum_{(u,v) \in E} \mathbf{1}_{\{0 < t - t_u - d_{uv} \leq \delta\}} w_{uv}(t - t_u - d_{uv}), \text{ where } t_u = \inf_{t \geq \min_{(z,u) \in E} \{t_z + d_{zu}\}} P_u(t) \geq \theta_u. \tag{4}$$

## 2.1 Computation in terms of firing time

Using (4) enables us to iteratively compute the firing time $t_v$ of each neuron $v \in V \setminus V_{\text{in}}$ if we know the firing time $t_u$ of each neuron $u \in V$ with $(u,v) \in E$ by solving for $t$ in

$$\inf_{t \geq \min_{(u,v) \in E} \{t_u + d_{uv}\}} P_v(t) = \inf_{t \geq \min_{(u,v) \in E} \{t_u + d_{uv}\}} \sum_{(u,v) \in E} \mathbf{1}_{\{0 < t - t_u - d_{uv} \leq \delta\}} w_{uv}(t - t_u - d_{uv}) = \theta_v. \quad (5)$$

Set $E(\mathbf{t}_U) := \{(u,v) \in E : d_{uv} + t_u < t_v \leq d_{uv} + t_u + \delta\}$, where $\mathbf{t}_U := (t_u)_{(u,v) \in E}$ is a vector containing the given firing times of the presynaptic neurons. The firing time $t_v$ satisfies

$$\theta_v = \sum_{(u,v) \in E} \mathbf{1}_{\{0 < t - t_u - d_{uv} \leq \delta\}} w_{uv}(t_v - t_u - d_{uv}) = \sum_{(u,v) \in E(\mathbf{t}_U)} w_{uv}(t_v - t_u - d_{uv}), \quad (6)$$

$$\text{i.e., } t_v = \frac{\theta_v}{\sum_{(u,v) \in E(\mathbf{t}_U)} w_{uv}} + \frac{\sum_{(u,v) \in E(\mathbf{t}_U)} w_{uv}(t_u + d_{uv})}{\sum_{(u,v) \in E(\mathbf{t}_U)} w_{uv}}. \quad (7)$$

Here, $E(\mathbf{t}_U)$ identifies the presynaptic neurons that actually have an effect on $t_v$ based on $\mathbf{t}_U$. For instance, if $t_w > t_v$ for some synapse $(w,v) \in E$, then $w$ did not contribute to the firing of $v$ since the spike from $w$ arrived after $v$ already fired so that $(w,v) \notin E(\mathbf{t}_U)$. Equation (7) shows that $t_v$ is a weighted sum (up to a positive constant) of the firing times of neurons $u$ with $(u,v) \in E(\mathbf{t}_U)$. Flexibility, i.e., non-linearity, in this model is provided through the variation of the set $E(\mathbf{t}_U)$. Depending on the firing time of the presynaptic neurons $\mathbf{t}_U$ and the associated parameters (weights, delays, threshold), $E(\mathbf{t}_U)$ contains a set of different synapses so that $t_v$ via (7) alters accordingly.

We formally define SNNs and ANNs by a sequence of their parameters and their corresponding realizations in Appendix A.1. To employ an SNN, the (typically analog) input information needs to be encoded in the firing times of the neurons in the input layer, and similarly, the firing times of the output neurons need to be translated back to an appropriate target domain. The encoding scheme in Definition 3 in Appendix A.1 translates analog information into firing times and vice versa in a continuous manner. Note that the following results are valid within the aforementioned setting.

## 3 Main results

A broad class of ANNs based on a wide range of activation functions such as ReLU generate **C**ontinuous **P**iecewise **L**inear (CPWL) mappings ([6], [5]). In other words, these ANNs partition the input domain into regions, the so-called linear regions, on which an affine function represents the ANN's realization. The result in Theorem 1 shows that SNNs also express CPWL mappings under very general conditions.

**Theorem 1.** *Any SNN $\Phi$ realizes a CPWL function provided that the sum of synaptic weights of each neuron is positive and the encoding scheme is a CPWL function.*

*Proof.* We show in the Appendix (see Theorem 5) that the firing time of a spiking neuron with arbitrarily many input neurons is a CPWL function with respect to the input under the assumption that the sum of its weight is positive. Since $\Phi$ consists of spiking neurons arranged in layers it immediately follows that each layer realizes a CPWL mapping. Thus, as a composition of CPWL mappings, $\Phi$ itself realizes a CPWL function provided that the input and output encoding are also CPWL functions. $\qquad \square$

Next, we show that an SNN has the capacity to effectively reproduce the output of any (ReLU) ANN. In order to accurately realize the output of a ReLU network, the initial step involves realizing the ReLU activation function. Despite the fact that ReLU is a very basic CPWL function, we remark that it is not straightforward to realize ReLU via SNNs.

**Theorem 2.** *Let $a < 0 < b$. There does not exist a one-layer SNN that realizes $\sigma(x) = \max(0, x)$ on $[a, b]$. However, $\sigma$ can be realized by a two-layer SNN on $[a, b]$.*

The proof is constructive, and we refer to Appendix A.4 for a detailed proof. Next, we extend the realization of a ReLU neuron to the entire network. We only provide a short proof sketch; the details are deferred to the Appendix A.5.

**Theorem 3.** *Let $L, d \in \mathbb{N}$, $[a,b]^d \subset \mathbb{R}^d$ and let $\Psi$ be an arbitrary ANN of depth $L$ and fixed width $d$ employing a ReLU non-linearity, and having a one-dimensional output. Then, there exists an SNN $\Phi$ with $N(\Phi) = N(\Psi) + L(2d+3) - (2d+2)$ and $L(\Phi) = 3L - 2$ that realizes $\mathcal{R}_\Psi$ on $[a,b]^d$.*

*Sketch of proof.* Any multi-layer ANN with ReLU activation is simply an alternating composition of affine-linear functions and a non-linear function represented by ReLU. To realize the mapping generated by some arbitrary ANN, it suffices to realize the composition of affine-linear functions and the ReLU non-linearity and then extend the construction to the whole network using concatenation and parallelization operations defined in Appendix A.2. □

The aforementioned result can be generalized to ANNs with varying widths that employ any type of piecewise linear activation function. Our expressivity result in Theorem 3 implies that SNNs can essentially approximate any function with the same accuracy and (asymptotic) complexity bounds as (deep) ANNs employing a piecewise linear activation function, given the response function satisfies the introduced basic assumptions. The number of linear regions is another measure of expressivity that describes how well a neural network can fit a family of functions. The following result establishes the number of linear regions generated by a one-layer SNN.

**Theorem 4.** *Let $\Phi$ be a one-layer SNN with a single output neuron $v$ and $d$ input neurons $u_1, \ldots, u_d$ such that $\sum_{i=1}^{d} w_{u_i v} > 0$. Then $\Phi$ partitions the input domain into at most $2^d - 1$ linear regions. In particular, for a sufficiently large input domain, the maximal number of linear regions is attained if and only if all synaptic weights are positive.*

*Proof.* The maximum number of regions directly corresponds to $E(\mathbf{t}_U)$ defined in (7). Recall that $E(\mathbf{t}_U)$ identifies the presynaptic neurons that based on their firing times $\mathbf{t}_U = (t_{u_i})_{i=1}^{d}$ triggered the firing of $v$ at time $t_v$. Therefore, each region in the input domain is associated to a subset of input neurons that is responsible for the firing of $v$ on this specific domain. Hence, the number of regions is bounded by the number of non-empty subsets of $\{u_1, \ldots, u_d\}$, i.e., $2^d - 1$. Now, observe that any subset of input neurons can cause a spike in $v$ if and only if the sum of their weights is positive. Otherwise, the corresponding input region either does not exist or inputs from the corresponding region do not trigger a spike in $v$ since they can not increase the potential $P_v(t)$ as their net contribution is negative, i.e., the potential does not reach the threshold $\theta_v$. Hence, the maximal number of regions is attained if and only if all weights are positive and thereby the sum of weights of any subset of input neurons is positive as well. □

One-layer ReLU-ANNs and one-layer SNNs with one output neuron both partition the input domain into linear regions. A one-layer ReLU-ANN will partition the input domain into at most two linear regions, independent of the dimension of the input. In contrast, for a one-layer SNN, the maximum number of linear regions scales exponentially in the input dimension. This distinct behaviour stems from the intrinsic non-linearity of SNNs, originating from the subset of neurons affecting the output neuron's firing time, while in ANNs a non-linear function is applied to the output of neurons. Our result in Theorem 4 suggests that a shallow SNN can be as expressive as a deep ReLU network in terms of the number of linear regions required to express certain types of CPWL functions.

## 4 Discussion

The central aim of this paper is to study and compare the expressive power of SNNs and ANNs employing any piecewise linear activation function. The imperative role of time in biological neural systems accounts for differences in computation between SNNs and ANNs. The key difference in the realization of arbitrary CPWL mappings is the necessary size and complexity of the respective ANN and SNN. Recall that realizing the ReLU activation via SNNs required more computational units than the corresponding ANN (see Theorem 2). Conversely, using SNNs (see Theorem 4), one can also realize certain CPWL functions with fewer number of computational units and layers compared to ReLU-based ANNs. While neither model is clearly beneficial in terms of network complexity to express all CPWL functions, each model has distinct advantages and disadvantages. The significance of our results lies in investigating theoretically the approximation and expressivity capabilities of SNNs, highlighting their potential as an alternative computational model for complex tasks. The insights obtained from this work can further aid in designing architectures that can be implemented on neuromorphic hardware for energy-efficient applications.

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

# A  Appendix

**Outline**  We start by defining spiking and artificial neural networks and encoding scheme used in Section A.1. Subsequently, we introduce the spiking network calculus in Section A.2 to compose and parallelize different networks. In Section A.3, we provide the proof of Theorem 5. The proof of Theorem 2 is given in Section A.4. Finally, in Section A.5, we prove that an SNN can realize the output of any ReLU network.

## A.1  Input and output encoding

By restricting our framework of SNNs to acyclic graphs, we can arrange the underlying graph in layers and equivalently represent SNNs by a sequence of their parameters. This is analogous to the common representation of feedforward ANNs via a sequence of matrix-vector tuples [1], [20].

**Definition 2.** *Let $L \in \mathbb{N}$. A* spiking neural network $\Phi$ *associated to the acyclic graph $(V, E)$ is a sequence of matrix-matrix-vector tuples*

$$\Phi = ((W^1, D^1, \Theta^1), (W^2, D^2, \Theta^2), \ldots, (W^L, D^L, \Theta^L))$$

*where $N_0, \ldots, N_L \in \mathbb{N}$ and each $W^l \in \mathbb{R}^{N_{l-1} \times N_l}$, $D^l \in \mathbb{R}_+^{N_{l-1} \times N_l}$, and $\Theta^l \in \mathbb{R}_+^{N_l}$. The matrix entries $W_{uv}^l$ and $D_{uv}^l$ represent the weight and delay value associated with the synapse $(u, v) \in E$, respectively, and the entry $\Theta_v^l$ is the firing threshold associated with node $v \in V$. $N_0$ is the input dimension and $N_L$ is the output dimension of $\Phi$. We call $N(\Phi) := \sum_{j=0}^{L} N_j$ the number of neurons and $L(\Phi) := L$ denotes the number of layers of $\Phi$.*

**Remark 1.** *In an ANN, the input signal is propagated in a synchronized manner layer-wise through the network (see Definition 5). In contrast, in an SNN, information is transmitted via spikes, where spikes from layer $l - 1$ affect the membrane potential of layer $l$ neurons, resulting in asynchronous propagation due to variable firing times among neurons.*

To employ an SNN, the (typically analog) input information needs to be encoded in the firing times of the neurons in the input layer, and similarly, the firing times of the output neurons need to be translated back to an appropriate target domain. We will refer to this process as input encoding and output decoding. The applied encoding scheme certainly depends on the specific task at hand and the potential power and suitability of different encoding schemes is a topic that warrants separate investigation on its own. Our focus in this work lies on exploring the intrinsic capabilities of SNNs, rather than the specifics of the encoding scheme. Thus, we can formulate some guiding principles

for establishing a reasonable encoding scheme. First, the firing times of input and output neurons should encode analog information in a consistent way so that different networks can be concatenated in a well-defined manner. This enables us to construct suitable subnetworks and combine them appropriately to solve more complex tasks. Second, in the extreme case, the encoding scheme might directly contain the solution to a problem, underscoring the need for a sufficiently simple and broadly applicable encoding scheme to avoid this.

**Definition 3.** *Let $[a,b]^d \subset \mathbb{R}^d$ and $\Phi$ be an SNN with input neurons $u_1, \ldots, u_d$ and output neurons $v_1, \ldots, v_n$. Fix reference times $T_{in} \in \mathbb{R}^d$ and $T_{out} \in \mathbb{R}^n$. For any $x \in [a,b]^d$, we set the firing times of the input neurons to $(t_{u_1}, \ldots, t_{u_d})^T = T_{in} + x$ and the corresponding firing times of the output neurons $(t_{v_1}, \ldots, t_{v_n})^T = T_{out} + y$, determined via (7), encode the target $y \in \mathbb{R}^n$.*

**Remark 2.** *A bounded input range ensures that appropriate reference times can be fixed. Note that the introduced encoding scheme translates analog information into input firing times in a continuous manner. Occasionally, we will point out the effect of adjusting the scheme and we will sometimes with a slight abuse of notation refer to $T_{in}, T_{out}$ as one-dimensional objects, i.e., $T_{in}, T_{out} \in \mathbb{R}$ which is justified if the corresponding vectors contain the same element in each dimension.*

Next, we distinguish between a network and the target function it realizes. A network is a structured set of weights, delays and thresholds as defined in Definition 2, and the target function it realizes is the result of the asynchronous propagation of spikes through the network.

**Definition 4.** *On $[a,b]^d \subset \mathbb{R}^d$, the realization of an SNN $\Phi$ with output neurons $v_1, \ldots, v_n$ and reference times $T_{in} \in \mathbb{R}^d$ and $T_{out} \in \mathbb{R}^n$, where $T_{out} > T_{in}$, is defined as the map $\mathcal{R}_\Phi : \mathbb{R}^d \to \mathbb{R}^n$,*

$$\mathcal{R}_\Phi(x) = -T_{out} + (t_{v_1}, \ldots, t_{v_n})^T.$$

Next, we give a corresponding definition of an ANN and its realization.

**Definition 5.** *Let $L \in \mathbb{N}$. An* artificial neural network $\Psi$ *is a sequence of matrix-vector tuples*

$$\Psi = ((W^1, B^1), (W^2, B^2), \ldots, (W^L, B^L)),$$

*where $N_0, \ldots, N_L \in \mathbb{N}$ and each $W^l \in \mathbb{R}^{N_{l-1} \times N_l}$ and $B^l \in \mathbb{R}^{N_l}$. $N_0$ and $N_L$ are the input and output dimension of $\Psi$. We call $N(\Psi) := \sum_{j=0}^{L} N_j$ the number of neurons of the network $\Psi$, $L(\Psi) := L$ the number of layers of $\Psi$ and $N_l$ the width of $\Psi$ in layer $l$. The* realization *of $\Psi$ with* component-wise *activation function $\sigma : \mathbb{R} \to \mathbb{R}$ is defined as the map $\mathcal{R}_\Psi : \mathbb{R}^{N_0} \to \mathbb{R}^{N_L}$, $\mathcal{R}_\Psi(x) = y_L$, where $y_L$ results from*

$$y_0 = x, \quad y_l = \sigma(W^l y_{l-1} + B^l), \ for \ l = 1, \ldots, L-1, \quad and \quad y_L = W^L y_{L-1} + B^L. \quad (8)$$

In the remainder, we always employ the ReLU activation function $\sigma(x) = \max(0, x)$. One can perform basic actions on neural networks such as concatenation and parallelization to construct larger networks from existing ones. Adapting a general approach for ANNs as defined in [1], [20], we formally introduce the concatenation and parallelization of networks of spiking neurons in the next Section A.2.

## A.2 Spiking neural network calculus

It can be observed from Definition 3 that both inputs and outputs of SNNs are encoded in a unified format. This characteristic is crucial for concatenating/parallelizing two spiking network architectures that further enable us to attain compositions/parallelizations of network realizations.

We operate in the following setting: Let $L_1, L_2, d_1, d_2, d_1', d_2' \in \mathbb{N}$. Consider two SNNs $\Phi_1, \Phi_2$ given by

$$\Phi_i = ((W_1^i, D_1^i, \Theta_1^i), \ldots, (W_{L_i}^i, D_{L_i}^i, \Theta_{L_i}^i)), \quad i = 1, 2,$$

with input domains $[a_1, b_1]^{d_1} \subset \mathbb{R}^{d_1}$, $[a_2, b_2]^{d_2} \subset \mathbb{R}^{d_2}$ and output dimension $d_1', d_2'$, respectively. Denote the input neurons by $u_1, \ldots, u_{d_i}$ with respective firing times $t_{u_j}^i$ and the output neurons by $v_1, \ldots, v_{d_i'}$ with respective firing times $t_{v_j}^i$ for $i = 1, 2$. By Definition 3, we can express the firing times of the input neurons as

$$t_u^1(x) := (t_{u_1}^1, \ldots, t_{u_{d_1}}^1)^T = T_{in}^1 + x \quad \text{for } x \in [a_1, b_1]^{d_1},$$

$$t_u^2(x) := (t_{u_1}^2, \ldots, t_{u_{d_2}}^2)^T = T_{in}^2 + x \quad \text{for } x \in [a_2, b_2]^{d_2} \quad (9)$$

and, by Definition 4, the realization of the networks as

$$\mathcal{R}_{\Phi_1}(x) = -T_{\text{out}}^1 + t_v^1(t_u^1(x)) := -T_{\text{out}}^1 + (t_{v_1}^1, \ldots, t_{v_{d_1'}}^1)^T \quad \text{for } x \in [a_1, b_1]^{d_1},$$

$$\mathcal{R}_{\Phi_2}(x) = -T_{\text{out}}^2 + t_v^2(t_u^2(x)) := -T_{\text{out}}^2 + (t_{v_1}^2, \ldots, t_{v_{d_2'}}^2)^T \quad \text{for } x \in [a_2, b_2]^{d_2} \qquad (10)$$

for some constants $T_{\text{in}}^1 \in \mathbb{R}^{d_1}$, $T_{\text{in}}^2 \in \mathbb{R}^{d_2}$, $T_{\text{out}}^1 \in \mathbb{R}^{d_1'}$, $T_{\text{out}}^2 \in \mathbb{R}^{d_2'}$.

We define the concatenation of the two networks in the following way.

**Definition 6.** *(Concatenation) Let $\Phi_1$ and $\Phi_2$ be such that the input layer of $\Phi_1$ has the same dimension as the output layer of $\Phi_2$, i.e., $d_2' = d_1$. Then, the concatenation of $\Phi_1$ and $\Phi_2$, denoted as $\Phi_1 \bullet \Phi_2$, represents the $(L_1 + L_2)$-layer network*

$$\Phi_1 \bullet \Phi_2 := ((W_1^2, D_1^2, \Theta_1^2), \ldots, (W_{L_2}^2, D_{L_2}^2, \Theta_{L_2}^2), (W_1^1, D_1^1, \Theta_1^1), \ldots, (W_{L_1}^1, D_{L_1}^1, \Theta_{L_1}^1)).$$

**Lemma 1.** *Let $d_2' = d_1$ and fix $T_{in} = T_{in}^2$ and $T_{out} = T_{out}^1$. If $T_{out}^2 = T_{in}^1$ and $\mathcal{R}_{\Phi_2}([a_2, b_2]^{d_2}) \subset [a_1, b_1]^{d_1}$, then*

$$\mathcal{R}_{\Phi_1 \bullet \Phi_2}(x) = \mathcal{R}_{\Phi_1}(\mathcal{R}_{\Phi_2}(x)) \quad \text{for all } x \in [a, b]^{d_2}$$

*with respect to the reference times $T_{in}, T_{out}$. Moreover, $\Phi_1 \bullet \Phi_2$ is composed of $N(\Phi_1) + N(\Phi_2) - d_1$ computational units.*

*Proof.* It is straightforward to verify via the construction that the network $\Phi_1 \bullet \Phi_2$ is composed of $N(\Phi_1) + N(\Phi_2) - d_1$ computational units. Moreover, under the given assumptions $\mathcal{R}_{\Phi_1} \circ \mathcal{R}_{\Phi_2}$ is well-defined so that (9) and (10) imply

$$\mathcal{R}_{\Phi_1 \bullet \Phi_2}(x) = -T_{\text{out}}^1 + t_v^1(t_v^2(T_{\text{in}} + x)) = -T_{\text{out}}^1 + t_v^1(t_v^2(T_{\text{in}}^2 + x)) = -T_{\text{out}}^1 + t_v^1(t_v^2(t_u^2(x)))$$

$$= -T_{\text{out}}^1 + t_v^1(T_{\text{out}}^2 + \mathcal{R}_{\Phi_2}(x)) = -T_{\text{out}}^1 + t_v^1(T_{\text{in}}^1 + \mathcal{R}_{\Phi_2}(x))$$

$$= -T_{\text{out}}^1 + t_v^1(t_u^1(\mathcal{R}_{\Phi_2}(x))) = \mathcal{R}_{\Phi_1}(\mathcal{R}_{\Phi_2}(x)) \quad \text{for } x \in [a_2, b_2]^{d_2}.$$

$\square$

In addition to concatenating networks, we also perform parallelization operation on SNNs.

**Definition 7.** *(Parallelization) Let $\Phi_1$ and $\Phi_2$ be such that they have the same depth and input dimension, i.e., $L_1 = L_2 =: L$ and $d_1 = d_2 =: d$. Then, the parallelization of $\Phi_1$ and $\Phi_2$, denoted as $P(\Phi_1, \Phi_2)$, represents the $L$-layer network with $d$-dimensional input*

$$P(\Phi_1, \Phi_2) := ((\tilde{W}_1, \tilde{D}_1, \tilde{\Theta}_1), \ldots, (\tilde{W}_L, \tilde{D}_L, \tilde{\Theta}_L)),$$

*where*

$$\tilde{W}_1 = \begin{pmatrix} W_1^1 & W_1^2 \end{pmatrix}, \quad \tilde{D}_1 = \begin{pmatrix} D_1^1 & D_1^2 \end{pmatrix}, \quad \tilde{\Theta}_1 = \begin{pmatrix} \Theta_1^1 \\ \Theta_1^2 \end{pmatrix}$$

*and*

$$\tilde{W}_l = \begin{pmatrix} W_l^1 & 0 \\ 0 & W_l^2 \end{pmatrix}, \quad \tilde{D}_l = \begin{pmatrix} D_l^1 & 0 \\ 0 & D_l^2 \end{pmatrix}, \quad \tilde{\Theta}_l = \begin{pmatrix} \Theta_l^1 \\ \Theta_l^2 \end{pmatrix}, \quad \text{for } 1 < l \leq L.$$

**Lemma 2.** *Let $d := d_2 = d_1$ and fix $T_{in} := T_{in}^1$, $T_{out} := (T_{out}^1, T_{out}^2)$, $a := a_1$ and $b := b_1$. If $T_{in}^2 = T_{in}^1$, $T_{out}^2 = T_{out}^1$ and $a_1 = a_2$, $b_1 = b_2$, then*

$$\mathcal{R}_{P(\Phi_1, \Phi_2)}(x) = (\mathcal{R}_{\Phi_1}(x), \mathcal{R}_{\Phi_2}(x)) \quad \text{for } x \in [a, b]^d$$

*with respect to the reference times $T_{in}, T_{out}$. Moreover, $P(\Phi_1, \Phi_2)$ is composed of $N(\Phi_1) + N(\Phi_2) - d$ computational units.*

*Proof.* The number of computational units is an immediate consequence of the construction. Since the input domains of $\Phi_1$ and $\Phi_2$ agree, (9) and (10) show that

$$\mathcal{R}_{P(\Phi_1, \Phi_2)}(x) = -T_{\text{out}} + (t_v^1(T_{\text{in}} + x), t_v^2(T_{\text{in}} + x)) = (-T_{\text{out}}^1 + t_v^1(T_{\text{in}}^1 + x), -T_{\text{out}}^2 + t_v^2(T_{\text{in}}^2 + x))$$

$$= (-T_{\text{out}}^1 + t_v^1(t_u^1(x)), -T_{\text{out}}^2 + t_v^2(t_u^2(x))) = (\mathcal{R}_{\Phi_1}(x), \mathcal{R}_{\Phi_2}(x)) \quad \text{for } x \in [a, b]^d.$$

$\square$

**Remark 3.** *Note that parallelization and concatenation can be straightforwardly extended (recursively) to a finite number of networks. Additionally, more general forms of parallelization and concatenations of networks, e.g., parallelization of networks with different depths, can be established. However, for the constructions presented in this work, the introduced notions suffice.*

## A.3 Realizations of spiking neural networks

In this section, we show that a spiking neuron generates a CPWL mapping.

**Theorem 5.** *Let $v$ be a spiking neuron with $d$ input neurons $u_1, \ldots, u_d$. The firing time $t_v(t_{u_1}, \ldots, t_{u_d})$ as a function of the firing times $t_{u_1}, \ldots, t_{u_d}$ is a CPWL mapping provided that $\sum_{i=1}^{d} w_{u_i v} > 0$, where $w_{u_i v} \in \mathbb{R}$ is the synaptic weight between $u_i$ and $v$.*

*Proof.* The condition $\sum_{i=1} w_{u_i v} > 0$ simply ensures that the input domain is decomposed into regions associated with subsets of input neurons with positive net weight. If $\sum_{i=1} w_{u_i v} < 0$, then the corresponding input region either does not exist or inputs from the corresponding region do not trigger a spike in $v$ since they can not increase the potential $P_v(t)$ as their net contribution is negative, i.e., the potential does not reach the threshold $\theta_v$. Hence, with $\sum_{i=1} w_{u_i v} > 0$, the situation described above can not arise and the notion of a CPWL mapping on $\mathbb{R}^d$ is well-defined. Denote the associated delays by $d_{u_i v} \geq 0$ and the threshold of $v$ by $\theta_v > 0$. We distinguish between the $2^d - 1$ variants of input combinations that can cause a firing of $v$. Let $I \subset \{1, \ldots, d\}$ be a non-empty subset and $I^c$ the complement of $I$ in $\{1, \ldots, d\}$, i.e., $I^c = \{1, \ldots, d\} \setminus I$. Assume that all $u_i$ with $i \in I$ contribute to the firing of $v$ whereas spikes from $u_i$ with $i \in I^c$ do not influence the firing of $v$. Then $\sum_{i \in I} w_{u_i v}$ is required to be positive, and by (6) and (7) the following holds:

$$t_{u_k} + d_{u_k v} \geq t_v = \frac{\theta_v}{\sum_{i \in I} w_{u_i v}} + \sum_{i \in I} \frac{w_{u_i v}}{\sum_{j \in I} w_{u_j v}}(t_{u_i} + d_{u_i v}) \quad \text{for all } k \in I^c \qquad (11)$$

and

$$t_{u_k} + d_{u_k v} < t_v = \frac{\theta_v}{\sum_{i \in I} w_{u_i v}} + \sum_{i \in I} \frac{w_{u_i v}}{\sum_{j \in I} w_{u_j v}}(t_{u_i} + d_{u_i v}) \quad \text{for all } k \in I. \qquad (12)$$

Rewriting yields

$$t_{u_k} \geq \frac{\theta_v}{\sum_{i \in I} w_{u_i v}} + \sum_{i \in I} \frac{w_{u_i v}}{\sum_{j \in I} w_{u_j v}}(t_{u_i} + d_{u_i v}) - d_{u_k v} \quad \text{for all } k \in I^c \qquad (13)$$

and

$$t_{u_k} \begin{cases} < \frac{\theta_v}{\sum_{j \in I \setminus k} w_{u_j v}} + \sum_{i \in I \setminus k} \frac{w_{u_i v}}{\sum_{j \in I \setminus k} w_{u_j v}}(t_{u_i} + d_{u_i v}) - d_{u_k v}, & \text{if } \frac{\sum_{i \in I \setminus k} w_{u_i v}}{\sum_{i \in I} w_{u_i v}} > 0 \\ > \frac{\theta_v}{\sum_{j \in I \setminus k} w_{u_j v}} + \sum_{i \in I \setminus k} \frac{w_{u_i v}}{\sum_{j \in I \setminus k} w_{u_j v}}(t_{u_i} + d_{u_i v}) - d_{u_k v}, & \text{if } \frac{\sum_{i \in I \setminus k} w_{u_i v}}{\sum_{i \in I} w_{u_i v}} < 0 \end{cases} \quad \forall k \in I.$$

It is now clear that the firing time $t_v(t_{u_1}, \ldots, t_{u_d})$ as a function of the input $t_{u_1}, \ldots, t_{u_d}$ is a piecewise linear mapping on polytopes decomposing $\mathbb{R}^d$. To show that the mapping is additionally continuous, we need to assess $t_v(t_{u_1}, \ldots, t_{u_d})$ on the breakpoints. Let $I, J \subset \{1, \ldots, d\}$ be index sets corresponding to input neurons $\{u_i : i \in I\}, \{u_j : j \in J\}$ that cause $v$ to fire on the input region $R^I \subset \mathbb{R}^d$, $R^J \subset \mathbb{R}^d$ respectively. Assume that it is possible to transition from $R^I$ to $R^J$ through a breakpoint $t^{I,J} = (t_{u_1}^{I,J}, \ldots, t_{u_d}^{I,J}) \in \mathbb{R}^d$ without leaving $R^I \cup R^J$. Crossing the breakpoint is equivalent to the fact that the input neurons $\{u_i : i \in I \setminus J\}$ do not contribute to the firing of $v$ anymore and the input neurons $\{u_i : i \in J \setminus I\}$ begin to contribute to the firing of $v$.

Assume first that $J \subset I$. Then, we observe that the breakpoint $t^{I,J}$ is necessarily an element of the linear region corresponding to the index set with smaller cardinality, i.e., $t^{I,J} \in R^J$. This is an immediate consequence of (12) and the fact that $t^{I,J}$ is characterized by

$$t_{u_k}^{I,J} + d_{u_k v} = t_v(t^{I,J}) \quad \text{for all } k \in I \setminus J. \qquad (14)$$

Indeed, if $t_{u_k}^{I,J} + d_{u_k v} > t_v(t^{I,J})$, then there exists $\varepsilon_k > 0$ such that (13) also holds for $t_{u_k}^{I,J} \pm \varepsilon$, where $0 \leq \varepsilon < \varepsilon_k$, i.e., a small change in $t_{u_k}^{I,J}$ is not sufficient to change the corresponding linear region, contradicting our assumption that $t^{I,J}$ is a breakpoint.

The firing time $t_v(t^{I,J})$ is explicitly given by

$$t_v(t^{I,J}) = \frac{\theta_v}{\sum_{i \in J} w_{u_i v}} + \sum_{i \in J} \frac{w_{u_i v}}{\sum_{j \in J} w_{u_j v}}(t_{u_i}^{I,J} + d_{u_i v})$$

Using (14), we obtain

$$0 = -\frac{w_{u_k v}}{\sum_{j \in J} w_{u_j v}}(t_v(t^{I,J}) - (t^{I,J}_{u_k} + d_{u_k v})) \quad \text{for all } k \in I \setminus J$$

so that

$$t_v(t^{I,J}) = \frac{\theta_v}{\sum_{i \in J} w_{u_i v}} + \sum_{i \in J} \frac{w_{u_i v}}{\sum_{j \in J} w_{u_j v}}(t^{I,J}_{u_i} + d_{u_i v}) - \sum_{i \in I \setminus J} \frac{w_{u_i v}}{\sum_{j \in J} w_{u_j v}}(t_v(t^{I,J}) - (t^{I,J}_{u_i} + d_{u_i v})).$$

Solving for $t_v(t^{I,J})$ yields

$$\begin{aligned}
t_v(t^{I,J}) &= \Big(1 + \sum_{i \in I \setminus J} \frac{w_{u_i v}}{\sum_{j \in J} w_{u_j v}}\Big)^{-1} \cdot \Big(\frac{\theta_v}{\sum_{i \in J} w_{u_i v}} + \sum_{i \in I} \frac{w_{u_i v}}{\sum_{j \in J} w_{u_j v}}(t^{I,J}_{u_i} + d_{u_i v})\Big) \\
&= \sum_{i \in J} \frac{w_{u_i v}}{\sum_{j \in I} w_{u_j v}} \cdot \Big(\frac{\theta_v}{\sum_{i \in J} w_{u_i v}} + \sum_{i \in I} \frac{w_{u_i v}}{\sum_{j \in J} w_{u_j v}}(t^{I,J}_{u_i} + d_{u_i v})\Big) \\
&= \frac{\theta_v}{\sum_{i \in I} w_{u_i v}} + \sum_{i \in I} \frac{w_{u_i v}}{\sum_{j \in I} w_{u_j v}}(t^{I,J}_{u_i} + d_{u_i v}),
\end{aligned}$$

which is exactly the expression for the firing time on $R^I$. This shows that $t_v(t_{u_1}, \dots, t_{u_d})$ is continuous in $t^{I,J}$. Since the breakpoint $t^{I,J}$ was chosen arbitrarily, $t_v(t_{u_1}, \dots, t_{u_d})$ is continuous at any breakpoint.

The case $I \subset J$ follows analogously. It remains to check the case when neither $I \subset J$ nor $J \subset I$. To that end, let $i^* \in I \setminus J$ and $j^* \in J \setminus I$. Assume without loss of generality that $t^{I,J} \in R^I$ so that (11) and (12) imply

$$t^{I,J}_{u_{i^*}} + d_{u_{i^*} v} < t_v(t^{I,J}) \le t^{I,J}_{u_{j^*}} + d_{u_{j^*} v}.$$

Hence, there exists $\varepsilon > 0$ such that

$$t^{I,J}_{u_{i^*}} + d_{u_{i^*} v} < t^{I,J}_{u_{j^*}} + d_{u_{j^*} v} - \varepsilon. \tag{15}$$

Moreover, due to the fact that $t^{I,J}$ is a breakpoint we can find $t^J \in R^J \cap \mathcal{B}(t^{I,J}; \frac{\varepsilon}{3})$, where $\mathcal{B}(t^{I,J}; \frac{\varepsilon}{3})$ denotes the open ball with radius $\frac{\varepsilon}{3}$ centered at $t^{I,J}$. In particular, this entails that

$$-\frac{\varepsilon}{3} < (t^J_{u_{i^*}} - t^{I,J}_{u_{i^*}}), (t^{I,J}_{u_{j^*}} - t^J_{u_{j^*}}) < \frac{\varepsilon}{3},$$

and therefore together with (15)

$$\begin{aligned}
t^J_{u_{i^*}} + d_{u_{i^*} v} - (t^J_{u_{j^*}} + d_{u_{j^*} v}) &= (t^J_{u_{i^*}} - t^{I,J}_{u_{i^*}}) + (t^{I,J}_{u_{i^*}} + d_{u_{i^*} v} - (t^{I,J}_{u_{j^*}} + d_{u_{j^*} v})) + (t^{I,J}_{u_{j^*}} - t^J_{u_{j^*}}) \\
&< 0, \quad \text{i.e., } t^J_{u_{i^*}} + d_{u_{i^*} v} < t^J_{u_{j^*}} + d_{u_{j^*} v}.
\end{aligned}$$

However, (11) and (12) require that

$$t^J_{u_{j^*}} + d_{u_{j^*} v} < t_v(t^J) \le t^J_{u_{i^*}} + d_{u_{i^*} v}$$

since $t^J \in R^J$. Thus, $t^{I,J}$ can not exist and the case when neither $I \subset J$ nor $J \subset I$ can not arise. $\quad\square$

## A.4 Realizing ReLU with spiking neural networks

**Proposition 1.** *Let $c_1 \in \mathbb{R}$, $c_2 \in (a, b) \subset \mathbb{R}$ and consider $f_1, f_2 : [a, b] \to \mathbb{R}$ defined as*

$$f_1(x) = \begin{cases} x + c_1 & , \text{if } x > c_2 \\ c_1 & , \text{if } x \le c_2 \end{cases} \quad \text{or} \quad f_2(x) = \begin{cases} x + c_1 & , \text{if } x < c_2 \\ c_1 & , \text{if } x \ge c_2 \end{cases}.$$

*There does not exist a one-layer SNN with output neuron $v$ and input neuron $u_1$ such that $t_v(x) = f_i(x)$, $i = 1, 2$, on $[a, b]$, where $t_v(x)$ denotes the firing time of $v$ on input $t_{u_1} = x$.*

*Proof.* First, note that a one-layer SNN realizes a CPWL function. For $c_2 \neq 0$, $f_i$ is not continuous and therefore can not be emulated by the firing time of any one-layer SNN. Hence, it is left to consider the case $c_2 = 0$. If $u_1$ is the only input neuron, then $v$ fires if and only if $w_{u_1 v} > 0$ and by (7) the firing time is given by

$$t_v(x) = \frac{\theta}{w_{u_1 v}} + x + d_{u_1 v} \quad \text{for all } x \in [a, b],$$

i.e., $t_v \neq f_i$. Therefore, we introduce auxiliary input neurons $u_2, \ldots, u_n$ and assume without loss of generality that $t_{u_i} + d_{u_i v} < t_{u_j} + d_{u_j v}$ for $j > i$. Here, the firing times $t_{u_i}$, $i = 2, \ldots, n$, are suitable constants. We will show that even in this extended setting $t_v \neq f_i$ still holds and thereby also the claim.

For the sake of contradiction, assume that $t_v(x) = f_1(x)$ for all $x \in [a, b]$. This implies that there exists an index set $J \subset \{1, \ldots, n\}$ with $\sum_{j \in J} w_{u_j v} > 0$ and a corresponding interval $(a_1, 0] \subset [a, b]$ such that

$$c_1 = t_v(x) = \frac{1}{\sum_{i \in J} w_{u_i v}} \left( \theta_v + \sum_{i \in J} w_{u_i v}(t_{u_i} + d_{u_i v}) \right) \quad \text{for all } x \in (a_1, 0],$$

where we have applied (7). Moreover, $J$ is of the form $J = \{2, \ldots, \ell\}$ for some $\ell \in \{1, \ldots, n\}$ because $(t_{u_i} + d_{u_i v})_{i=2}^n$ is in ascending order, i.e., if the spike from $u_\ell$ has reached $v$ before $v$ fired, then so did the spikes from $u_i$, $2 \leq i < \ell$. Additionally, we know that $1 \notin J$ since otherwise $t_v$ is non-constant on $(a_1, 0]$ (due to the contribution from $u_1$), i.e., $t_v \neq c_1$ on $(a_1, 0]$. In particular, the spike from $u_1$ reaches $v$ after the neurons $u_2, \ldots, u_\ell$ already caused $v$ to fire, i.e., we have

$$x + d_{u_1 v} \geq t_v(x) = c_1 \quad \text{for all } x \in (a_1, 0].$$

However, it immediately follows that

$$x + d_{u_1 v} > d_{u_1 v} \geq c_1 \quad \text{for all } x > 0.$$

Thus, we obtain $t_v(x) = c_1$ for $x > 0$ (since the spike from $u_1$ still reaches $v$ only after $v$ emitted a spike), which contradicts $t_v(x) = f_1(x)$ for all $x \in [a, b]$.

We perform a similar analysis to show that $f_2$ can not be emulated. For the sake of contradiction, assume that $t_v(x) = f_2(x)$ for all $x \in [a, b]$. This implies that there exists an index set $I \subset \{1, \ldots, n\}$ with $\sum_{i \in I} w_{u_i v} > 0$ and a corresponding interval $(a_2, 0) \subset [a, b]$ such that

$$x + c_1 = t_v(x) = \frac{1}{\sum_{i \in I} w_{u_i v}} \left( \theta_v + w_{u_1 v}(x + d_{u_1 v}) + \sum_{i \in I \setminus \{1\}} w_{u_i v}(t_{u_i} + d_{u_i v}) \right) \quad \text{for } x \in (a_2, 0),$$

(16)

where we have applied (7). We immediately observe that $1 \in I$, since otherwise $t_v$ is constant on $(a_2, 0)$. Moreover, by the same reasoning as before we can write $I = \{1, \ldots, \ell\}$ for some $\ell \in \{1, \ldots, n\}$. In order for $t_v(x) = f_2(x)$ for all $x \in [a, b]$ to hold, there needs to exist an index set $J \subset \{1, \ldots, n\}$ with $\sum_{j \in J} w_{u_j v} > 0$ and a corresponding interval $[0, b_2) \subset [a, b]$ such that $t_v = c_1$ on $[0, b_2)$. We conclude that $J = \{1, \ldots, m\}$ or $J = \{2, \ldots, m\}$ for some $m \in \{1, \ldots, n\}$. In the former case, $t_v$ is non-constant on $[0, b_2)$ (due to the contribution from $u_1$), i.e., $t_v \neq c_1$ on $[0, b_2)$. Hence, it remains to consider the latter case. Note that $m < \ell$ implies that $b_2 \leq a_2$ (as $u_2, \ldots, u_m$ already triggered a firing of $v$ before the spike from $u_\ell$ arrived) contradicting the construction $a_2 < 0 < b_2$. Similarly, $m = \ell$, i.e., $J = I \setminus \{1\}$ is not valid because (16) requires that

$$\frac{w_{u_1 v}}{\sum_{i \in I} w_{u_i v}} = 1 \Leftrightarrow \sum_{i \in I \setminus \{1\}} w_{u_i v} = 0 \Leftrightarrow \sum_{j \in J} w_{u_j v} = 0.$$

Finally, $m > \ell$ also results in a contradiction since

$$0 < \sum_{j \in J} w_{u_j v} = \sum_{i \in I \setminus \{1\}} w_{u_i v} + \sum_{j \in J \setminus I} w_{u_j v} = \sum_{j \in J \setminus I} w_{u_j v}$$

together with

$$0 < \sum_{i \in I} w_{u_i v} = \sum_{i \in I \setminus \{1\}} w_{u_i v} + w_{u_1 v} = w_{u_1 v}$$

imply that the neurons $\{u_j : j \in \{1\} \cup J\}$ also trigger a spike in $v$. However, the corresponding interval where the firing of $v$ is caused by $\{u_j : j \in \{1\} \cup J\}$ is necessarily located between $(a_2, 0)$ and $[0, b_2)$, which is not possible. $\square$

**Remark 4.** *The proof shows that* $-f_1$ *also can not be emulated by a one-layer SNN. Moreover, by adjusting (16) we observe that a necessary condition for* $-f_2$ *to be realized is that*

$$\frac{w_{u_1 v}}{\sum_{i \in I} w_{u_i v}} = -1 \Leftrightarrow - \sum_{i \in I \setminus \{1\}} w_{u_i v} = 2 w_{u_1 v} \Leftrightarrow -\frac{1}{2} \sum_{i \in I \setminus \{1\}} w_{u_i v} = w_{u_1 v}.$$

*Under this condition* $-f_2$ *can indeed be realized by a one-layer SNN as the following statement confirms.*

**Proposition 2.** *Let* $a < 0 < b, c$ *and consider* $f : [a, b] \to \mathbb{R}$ *defined as*

$$f(x) = \begin{cases} -x + c & , \text{ if } x < 0 \\ c & , \text{ if } x \geq 0 \end{cases}.$$

*There exists a one-layer SNN* $\Phi$ *with output neuron* $v$ *and input neuron* $u_1$ *such that* $t_v(x) = f(x)$ *on* $[a, b]$, *where* $t_v(x)$ *denotes the firing time of* $v$ *on input* $t_{u_1} = x$.

*Proof.* We introduce an auxiliary input neuron with constant firing time $t_{u_2} \in \mathbb{R}$ and specify the parameter of $\Phi = ((W, D, \Theta))$ in the following manner (see Figure 1a):

$$W = \begin{pmatrix} -\frac{1}{2} \\ 1 \end{pmatrix}, D = \begin{pmatrix} d_1 \\ d_2 \end{pmatrix}, \Theta = \theta,$$

where $\theta, d_1, d_2 > 0$ are to be specified. Note that either $u_2$ or $u_1$ together with $u_2$ can trigger a spike in $v$ since $w_{u_1 v} < 0$. Therefore, applying (7) yields that $u_2$ triggers a spike in $v$ under the following circumstances:

$$t_v(x) = \theta + t_{u_2} + d_2 \quad \text{if } t_v(x) \leq t_{u_1} + d_1 = x + d_1.$$

Hence, this case only arises when

$$\theta + t_{u_2} + d_2 \leq x + d_1 \Leftrightarrow \theta + t_{u_2} + d_2 - d_1 \leq x.$$

To emulate $f$ the parameter needs to satisfy

$$\theta + t_{u_2} + d_2 - d_1 \leq x \text{ for all } x \in [0, b] \quad \text{and} \quad \theta + t_{u_2} + d_2 - d_1 > x \text{ for all } x \in [a, 0)$$

which simplifies to

$$\theta + t_{u_2} + d_2 - d_1 = 0. \tag{17}$$

If the additional condition

$$\theta + t_{u_2} + d_2 = c \tag{18}$$

is met, we can infer that

$$t_v(x) = \begin{cases} 2(\theta + t_{u_2} + d_2) - (x + d_1) & , \text{ if } x < 0 \\ \theta + t_{u_2} + d_2 & , \text{ if } x \geq 0 \end{cases} = \begin{cases} -x + c & , \text{ if } x < 0 \\ c & , \text{ if } x \geq 0 \end{cases}.$$

Finally, it is immediate to verify that the conditions (17) and (18) can be satisfied simultaneously due to the assumption that $c > 0$, e.g., choosing $d_1 = d_2 = c$ and $t_{u_2} = -\theta$ is sufficient. $\square$

**Remark 5.** *We wish to mention that we can not adapt the previous construction to emulate ReLU with a consistent encoding scheme, i.e., such that the input and output firing times encode analog values in the same format with respect to reference times* $T_{in}, T_{out} \in \mathbb{R}$, $T_{in} < T_{out}$. *Indeed, it is obvious that using the input encoding* $T_{in} + x$ *and output decoding* $-T_{out} + t_v$, *does not realize ReLU. Similarly, one verifies that the input encoding* $T_{in} - x$ *and output decoding* $T_{out} - t_v$ *also does not yield the desired function. However, choosing the input encoding* $T_{in} - x$ *and output decoding* $-T_{out} + t_v$ *gives*

$$\mathcal{R}_\Phi(x) = \begin{cases} -T_{out} - T_{in} + c + x & , \text{ if } x > T_{in} \\ -T_{out} + c & , \text{ if } x \leq T_{in} \end{cases}.$$

*Setting* $T_{in} = 0$ *and* $T_{out} = c$ *implies that* $\Phi$ *realizes ReLU with inconsistent encoding* $T_{in} - x$ *and* $T_{out} + \mathcal{R}_\Phi(x)$. *Nevertheless, we want a consistent encoding scheme that allows us to compose ReLU (as typically is the case in ANNs) whereby an inconsistent scheme is disadvantageous.*

Applying the previous construction and adding another layer is adequate to emulate $f_1$ defined in Proposition 1 by a two-layer SNN.

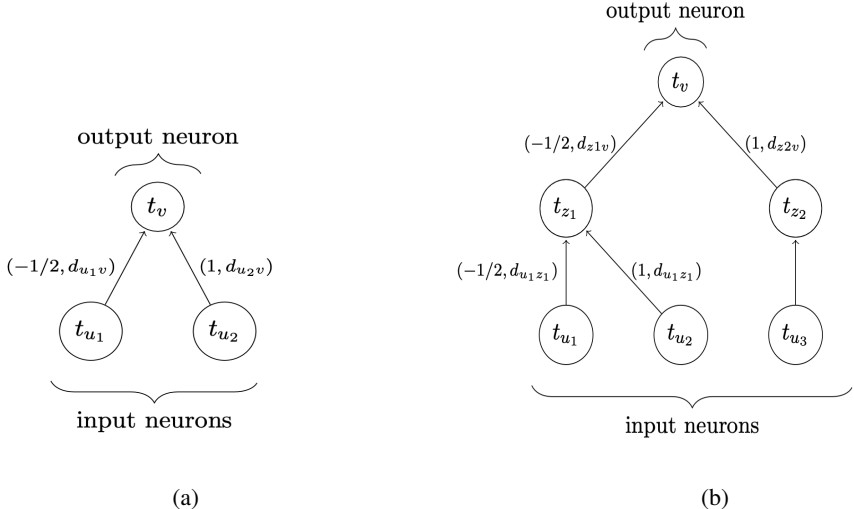

(a)                                             (b)

Figure 1: (a) Computation graph associated with a spiking network with two input neurons and one output neuron that realizes $f$ as defined in Proposition 2. (b) Stacking the network in (a) twice results in a spiking network that realizes the ReLU activation function.

**Proposition 3.** *Let $a < 0 < b < 0.5 \cdot c$ and consider $f : [a, b] \to \mathbb{R}$ defined as*

$$f(x) = \begin{cases} x + c & , \text{ if } x > 0 \\ c & , \text{ if } x \leq 0 \end{cases}$$

*There exists a 2-layer SNN $\Phi$ with output neuron $v$ and input neuron $u_1$ such that $t_v(x) = f(x)$ on $[a, b]$, where $t_v(x)$ denotes the firing time of $v$ on input $t_{u_1} = x$.*

*Proof.* We introduce an auxiliary input neuron $u_2$ with constant firing time $t_{u_2} \in \mathbb{R}$ and specify the parameter of $\Phi = ((W^1, D^1, \Theta^1), (W^2, D^2, \Theta^2))$ in the following manner:

$$W^1 = \begin{pmatrix} -\frac{1}{2} & 0 \\ 1 & 2 \end{pmatrix}, D^1 = \begin{pmatrix} d & 0 \\ d & \frac{d}{2} \end{pmatrix}, \Theta^1 = \begin{pmatrix} \theta \\ 2\theta \end{pmatrix}, W^2 = \begin{pmatrix} -\frac{1}{2} \\ 1 \end{pmatrix}, D^2 = \begin{pmatrix} d \\ d \end{pmatrix}, \Theta^2 = \theta, \quad (19)$$

where $d \geq 0$ and $\theta > 0$ is chosen such that $\theta + t_{u_2} > b$. We denote the input neurons by $u_1, u_2$, the neurons in the hidden layer by $z_1, z_2$ and the output neuron by $v$. Note that the firing time of $z_1$ depends on $u_1$ and $u_2$. In particular, either $u_2$ or $u_1$ together with $u_2$ can trigger a spike in $z_1$ since $w_{u_1 z_1} < 0$. Therefore, applying (7) yields that $u_2$ triggers a spike in $z_1$ under the following circumstances:

$$t_{z_1}(x) = \theta + t_{u_2} + d \quad \text{if } t_{z_1}(x) \leq t_{u_1} + d = x + d.$$

Hence, this case only arises when

$$\theta + t_{u_2} + d \leq x + d \Leftrightarrow \theta + t_{u_2} \leq x. \tag{20}$$

However, by construction $\theta + t_{u_2} > b$, so that (20) does not hold for any $x \in [a, b]$. Thus, we conclude via (7) that

$$t_{z_1}(x) = 2(\theta + t_{u_2} + d) - (x + d) = 2(\theta + t_{u_2}) + d - x.$$

By construction, the firing time $t_{z_2} = \theta + 2t_{u_2} + d$ of $z_2$ is a constant which depends on the input only via $u_2$. A similar analysis as in the first layer shows that

$$t_v(x) = \theta + t_{z_2} + d \quad \text{if } t_v(x) \leq t_{z_1} + d = 2(\theta + t_{u_2}) + d - x + d = 2(\theta + t_{u_2} + d) - x.$$

Hence, $z_2$ triggers a spike in $v$ when

$$\theta + \theta + 2t_{u_2} + d + d \leq 2(\theta + t_{u_2} + d) - x \quad \Leftrightarrow \quad x \leq 0.$$

If the additional condition

$$\theta + t_{z_2} + d = c \quad \Leftrightarrow \quad 2(\theta + d + t_{u_2}) = c \tag{21}$$

is met, we can infer that

$$t_v(x) = \begin{cases} 2(\theta + t_{z_2} + d) - (t_{z_1}(x) + d) & , \text{if } x > 0 \\ \theta + t_{z_2} + d & , \text{if } x \le 0 \end{cases}$$

$$= \begin{cases} 2c - (2(\theta + t_{u_2}) + d - x + d) & , \text{if } x > 0 \\ c & , \text{if } x \le 0 \end{cases}$$

$$= \begin{cases} x + c & , \text{if } x > 0 \\ c & , \text{if } x \le 0 \end{cases}.$$

Choosing $\theta$, $t_{u_2}$ and $d$ sufficiently small under the given constraints guarantees that (21) holds, i.e., $\Phi$ emulates $f$ as desired. $\qquad\square$

**Remark 6.** *It is again important to specify the encoding scheme via reference times $T_{in}, T_{out} \in \mathbb{R}$, $T_{in} < T_{out}$ to ensure that $\Phi$ realizes ReLU. The input encoding $T_{in} - x$ and output decoding $T_{out} - t_v$ does not yield the desired output since it results in a realization of the type $-ReLU(-x)$. In contrast, the input encoding $T_{in} + x$ and output decoding $-T_{out} + t_v$ with $T_{in} = 0$ and $T_{out} = c$ gives*

$$\mathcal{R}_\Phi(x) = -T_{out} + t_v(T_{in} + x) = -T_{out} + f(T_{in} + x) = \begin{cases} x & , \text{if } x > 0 \\ 0 & , \text{if } x \le 0 \end{cases} = ReLU(x).$$

*In this case, it is necessary to choose the reference time $T_{in} = 0$ to ensure that the breakpoint is also at zero. Next, we show that there is actually more freedom in choosing the reference time by analysing the construction in the proof more carefully.*

**Proposition 4.** *Let $a < 0 < b$ and consider $f : [a, b] \to \mathbb{R}$ defined as*

$$f(x) = \begin{cases} x & , \text{if } x > 0 \\ 0 & , \text{if } x \le 0 \end{cases}$$

*There exists a 2-layer SNN $\Phi$ with realization $\mathcal{R}_\Phi = f$ on $[a, b]$ with encoding scheme $T_{in} + x$ and decoding $-T_{out} + t_v$, where $v$ is the output neuron of $\Phi$, $T_{in} \in \mathbb{R}$ and $T_{out} = T_{in} + c$ for some constant $c > 0$ depending on the parameters of $\Phi$.*

*Proof.* Performing a similar construction with the following changes and the same analysis as in the proof of Proposition 3 yields the claim. First, we slightly adjust $\Phi = ((W^1, D^1, \Theta^1), (W^2, D^2, \Theta^2))$ in comparison to (19) and consider the network

$$W^1 = \begin{pmatrix} -\frac{1}{2} & 0 \\ 1 & 1 \end{pmatrix}, D^1 = \begin{pmatrix} d & 0 \\ d & d \end{pmatrix}, \Theta^1 = \begin{pmatrix} \theta \\ \theta \end{pmatrix}, W^2 = \begin{pmatrix} -\frac{1}{2} \\ 1 \end{pmatrix}, D^2 = \begin{pmatrix} d \\ d \end{pmatrix}, \Theta^2 = \theta,$$

where $d \ge 0$ and $\theta > b$ are fixed (see Figure 1b). Second, we choose the input reference time $T_{in} \in \mathbb{R}$ and fix the input of the auxiliary input neuron $u_2$ as $t_{u_2} = T_{in} \in \mathbb{R}$. Finally, setting the output reference time $T_{out} = 2(\theta + d) + T_{in}$ is sufficient to guarantee that $\Phi$ realizes $f$ on $[a, b]$. $\qquad\square$

### A.5    Realizing ReLU networks by spiking neural networks

In this section, we show that an SNN has the capability to reproduce the output of any ReLU network. Specifically, given access to the weights and biases of an ANN, we construct an SNN and set the parameter values based on the weights and biases of the given ANN. This leads us to the desired result. The essential part of our proof revolves around choosing the parameters of an SNN such that it effectively realizes the composition of an affine-linear map and the non-linearity represented by the ReLU activation. The realization of ReLU with SNNs is proved in the previous Section A.4. To realize an affine-linear function using a spiking neuron, it is necessary to ensure that the spikes from all the input neurons together result in the firing of an output neuron instead of any subset of the input neurons. We achieve that by appropriately adjusting the value of the threshold parameter. As a result, a spiking neuron, which implements an affine-linear map, avoids partitioning of the input space.

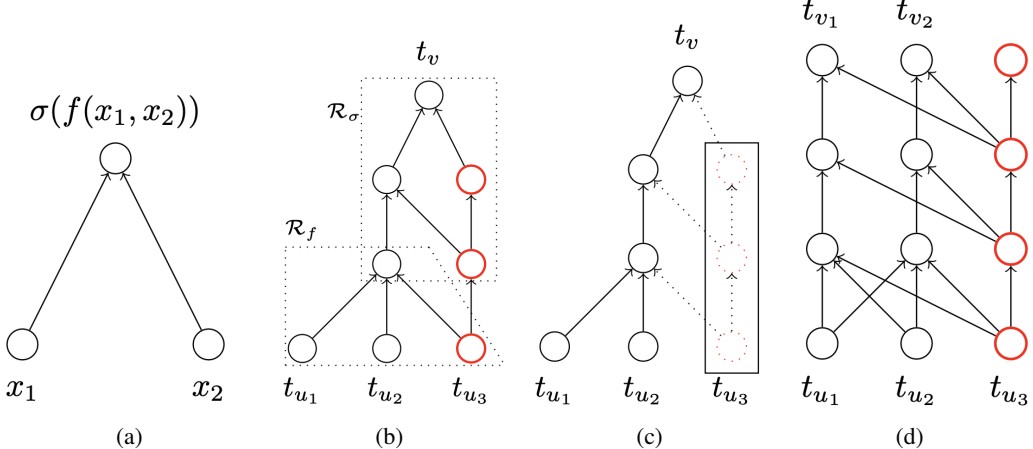

Figure 2: (a) Computation graph of an ANN with two input and one output unit realizing $\sigma(f(x_1, x_2))$, where $\sigma$ is the ReLU activation function. (b) Computation graph associated with an SNN resulting from the concatenation of $\Phi^\sigma$ and $\Phi^f$ that realizes $\sigma(f(x_1, x_2))$. The auxiliary neurons are shown in red. (c) Same computation graph as in (b); when parallelizing two identical networks, the dotted auxiliary neurons can be removed and auxiliary neurons from (b) can be used for each network instead. (d) Computation graph associated with a spiking network as a result of the parallelization of two subnetworks $\Phi^{\sigma \circ f_1}$ and $\Phi^{\sigma \circ f_2}$. The auxiliary neuron in the output layer serves the same purpose as the auxiliary neuron in the input layer and is needed when concatenating two such subnetworks $\Phi_{\sigma \circ f}$.

**Setup for the proof of Theorem 3**  Let $d, L \in \mathbb{N}$ be the width and the depth of an ANN $\Psi$, respectively, i.e.,

$$\Psi = ((A^1, B^1), (A^2, B^2), \ldots, (A^L, B^L)), \text{ where } (A^\ell, B^\ell) \in \mathbb{R}^{d \times d} \times \mathbb{R}^d, 1 \leq \ell < L,$$
$$(A^L, B^L) \in \mathbb{R}^{1 \times d} \times \mathbb{R}.$$

For a given input domain $[a, b]^d \subset \mathbb{R}^d$, we denote by $\Psi^\ell = ((A^\ell, B^\ell))$ the $\ell$-th layer, where $y^0 \in [a, b]^d$ and

$$y^l = \mathcal{R}_{\Psi^l}(y^{l-1}) = \sigma(A^l y^{l-1} + B^l), 1 \leq \ell < L,$$
$$y^L = \mathcal{R}_{\Psi^L}(y^{L-1}) = A^L y^{L-1} + B^L \tag{22}$$

so that $\mathcal{R}_\Psi = \mathcal{R}_{\Psi^L} \circ \cdots \circ \mathcal{R}_{\Psi^1}$.

For the construction of the corresponding SNN we refer to the associated weights and delays between two spiking neurons $u$ and $v$ by $w_{uv}$ and $d_{uv}$, respectively.

***Proof of Theorem 3***.  Any multi-layer ANN $\Psi$ with ReLU activation is simply an alternating composition of affine-linear functions $A^l y^{l-1} + B^l$ and a non-linear function represented by $\sigma$. To generate the mapping realized by $\Psi$, it suffices to realize the composition of affine-linear functions and the ReLU non-linearity and then extend the construction to the whole network using concatenation and parallelization operations. We prove the result via the following steps; see also Figure 2 for a depiction of the intermediate constructions.

**Step 1:** Realizing ReLU non-linearity.
Proposition 4 gives the desired result.

**Step 2:** Realizing affine-linear functions with one-dimensional range.
Let $f : [a, b]^d \to \mathbb{R}$ be an affine-linear function

$$f(x) = C^T x + s, \quad C^T = (c_1, \ldots, c_d) \in \mathbb{R}^d, s \in \mathbb{R}. \tag{23}$$

Consider a one-layer SNN that consists of an output neuron $v$ and d input units $u_1, \ldots, u_d$. Via (7) the firing time of $v$ as a function of the input firing times on the linear region $R^I$ corresponding to the

index set $I = \{1, \dots, d\}$ is given by

$$t_v(t_{u_1}, \dots, t_{u_d}) = \frac{\theta_v}{\sum_{i \in I} w_{u_i v}} + \frac{\sum_{i \in I} w_{u_i v}(t_{u_i} + d_{u_i v})}{\sum_{i \in I} w_{u_i v}} \quad \text{provided that} \sum_{i \in I} w_{u_i v} > 0.$$

Introducing an auxiliary input neuron $u_{d+1}$ with weight $w_{u_{d+1} v} = 1 - \sum_{i \in I} w_{u_i v}$ ensures that $\sum_{i \in I \cup \{d+1\}} w_{u_i v} > 0$ and leads to the firing time

$$t_v(t_{u_1}, \dots, t_{u_{d+1}}) = \theta_v + \sum_{i \in I \cup \{d+1\}} w_{u_i v}(t_{u_i} + d_{u_i v}) \quad \text{on } R^{I \cup \{d+1\}}.$$

Setting $w_{u_i v} = c_i$ for $i \in I$ and $d_{u_j v} = d' \geq 0$ for $j \in I \cup \{d+1\}$ yields

$$t_v(t_{u_1}, \dots, t_{u_{d+1}}) = \theta_v + w_{u_{d+1} v} \cdot t_{u_{d+1}} + d' + \sum_{i \in I} c_i t_{u_i} \text{ on } R^{I \cup \{d+1\}} \cap [a, b]^d.$$

Therefore, an SNN $\Phi^f = (W, D, \Theta)$ with parameters

$$W = \begin{pmatrix} c_1 \\ \vdots \\ c_{d+1} \end{pmatrix}, D = \begin{pmatrix} d' \\ \vdots \\ d' \end{pmatrix}, \Theta = \theta > 0, \quad \text{where } c_{d+1} = 1 - \sum_{i \in I} c_i,$$

and the usual encoding scheme $T_{\text{in}}/T_{\text{out}} + \cdot$ and fixed firing time $t_{u_{d+1}} = T_{\text{in}} \in \mathbb{R}$ realizes

$$\mathcal{R}_{\Phi^f}(x) = -T_{\text{out}} + t_v(T_{\text{in}} + x_1, \dots, T_{\text{in}} + x_d, T_{\text{in}}) = -T_{\text{out}} + \theta + T_{\text{in}} + d' + \sum_{i \in I} c_i x_i \quad (24)$$

$$= -T_{\text{out}} + \theta + T_{\text{in}} + d' + f(x_1, \dots, x_d) - s \quad \text{on } R^{I \cup \{d+1\}} \cap [a, b]^d. \quad (25)$$

Choosing a large enough threshold $\theta$ ensures that a spike in $v$ is necessarily triggered after all the spikes from $u_1, \dots, u_{d+1}$ reached $v$ so that $[a, b]^d \subset R^{I \cup \{d+1\}}$ holds. It suffices to set

$$\theta \geq \sup_{x \in [a,b]^d} \sup_{x_{\min} \leq t - T_{\text{in}} - d' \leq x_{\max}} P_v(t),$$

where $x_{\min} = \min\{x_1, \dots, x_d, 0\}$ and $x_{\max} = \max\{x_1, \dots, x_d, 0\}$, since this implies that the potential $P_v(t)$ is smaller than the threshold to trigger a spike in $v$ on the time interval associated to feasible input spikes, i.e., $v$ emits a spike after the last spike from an input neuron arrived at $v$. Applying (5) shows that for $x \in [a, b]^d$ and $t \in [x_{\min} + T_{\text{in}} + d', x_{\max} + T_{\text{in}} + d']$

$$P_v(t) = \sum_{i \in I} w_{u_i v}(t - (T_{\text{in}} + x_i) - d_{u_i v}) + w_{u_{d+1} v}(t - T_{\text{in}} - d_{u_{d+1} v}) = t - d' - T_{\text{in}} + \sum_{i \in I} c_i x_i$$

$$\leq x_{\max} + d \|C\|_\infty \|x\|_\infty \leq (1 + d \|C\|_\infty) \max\{|a|, |b|\}.$$

Hence, we set

$$\theta = (1 + d \|C\|_\infty) \max\{|a|, |b|\} + s + |s| \quad \text{and} \quad T_{\text{out}} = \theta - s + T_{\text{in}} + d'$$

to obtain via (24) that

$$\mathcal{R}_{\Phi^f}(x) = -T_{\text{out}} + t_v(T_{\text{in}} + x_1, \dots, T_{\text{in}} + x_d, T_{\text{in}}) = f(x) \quad \text{for } x \in [a, b]^d. \quad (26)$$

Note that the reference time $T_{\text{out}} = (1 + d \|C\|_\infty) \max\{|a|, |b|\} + |s| + T_{\text{in}} + d'$ is independent of the specific parameters of $f$ in the sense that only upper bounds $\|C\|_\infty, |s|$ on the parameters are relevant. Therefore, $T_{\text{out}}$ (with the associated choice of $\theta$) can be applied for different affine linear functions as long as the upper bounds remain valid. This is necessary for the composition and parallelization of subnetworks in the subsequent construction.

**Step 3:** Realizing compositions of affine-linear functions with one-dimensional range and ReLU.
The next step is to realize the composition of ReLU $\sigma$ with an affine linear mapping $f$ defined in (23). To that end, we want to concatenate the networks $\Phi^\sigma$ and $\Phi^f$ constructed in Step 1 and Step 2, respectively, via Lemma 1. To employ the concatenation operation we need to perform the following steps:

1. Find an appropriate input domain $[a', b'] \subset \mathbb{R}$, that contains the image $f([a,b]^d)$ so that parameters and reference times of $\Phi^\sigma$ can be fixed appropriately (see Proposition 4 for the detailed conditions on how to choose the parameter).

2. Ensure that the output reference time $T_{\text{out}}^f$ of $\Phi^f$ equals the input reference time $T_{\text{in}}^\sigma$ of $\Phi^\sigma$.

3. Ensure that the number of neurons in the output layer of $\Phi^f$ is the same as the number of input neurons in $\Phi^\sigma$.

For the first point, note that

$$|f(x)| = |C^T x + s| \leq d \, \|C\|_\infty \cdot \|x\|_\infty + |s| \leq d \, \|C\|_\infty \cdot \max\{|a|, |b|\} + |s| \text{ for all } x \in [a,b]^d.$$

Hence, we can use the input domain

$$[a', b'] = [-d \, \|C\|_\infty \cdot \max\{|a|, |b|\} + |s|, d \, \|C\|_\infty \cdot \max\{|a|, |b|\} + |s|]$$

and specify the parameters of $\Phi^\sigma$ accordingly. Additionally, recall from Proposition 4 that $T_{\text{in}}^\sigma$ can be chosen freely, so we may fix $T_{\text{in}}^\sigma = T_{\text{out}}^f$, where $T_{\text{out}}^f$ is established in Step 2. It remains to consider the third point. In order to realize ReLU an additional auxiliary neuron in the input layer of $\Phi^\sigma$ with constant input $T_{\text{in}}^\sigma$ was introduced. Hence, we also need to add an additional output neuron in $\Phi^f$ with (constant) firing time $T_{\text{out}}^f = T_{\text{in}}^\sigma$ so that the corresponding output and input dimension and their specification match. This is achieved by introducing a single synapse from the auxiliary neuron in the input layer of $\Phi^f$ to the newly added output neuron and by specifying the parameters of the newly introduced synapse and neuron suitably. Formally, the adapted network $\Phi^f = (W, D, \Theta)$ is given by

$$W = \begin{pmatrix} c_1 & 0 \\ \vdots & \vdots \\ c_d & 0 \\ c_{d+1} & 1 \end{pmatrix}, D = \begin{pmatrix} d' & 0 \\ \vdots & \vdots \\ d' & 0 \\ d' & d' \end{pmatrix}, \Theta = \begin{pmatrix} \theta \\ T_{\text{out}}^f - T_{\text{in}}^f - d' \end{pmatrix},$$

where the values of the parameters are specified in Step 2.

Then the realization of the concatenated network $\Phi^{\sigma \circ f}$ is the composition of the individual realizations. This is exemplarily demonstrated in Figure 2b for the two-dimensional input case. By analyzing $\Phi^{\sigma \circ f}$, we conclude that a three-layer SNN with

$$N(\Phi^{\sigma \circ f}) = N(\Phi^\sigma) - N_0(\Phi^\sigma) + N(\Phi^f) = 5 - 2 + d + 3 = d + 6$$

computational units can realize $\sigma \circ f$ on $[a,b]^d$, where $N_0(\Phi^\sigma)$ denotes the number of neurons in the input layer of $\Phi^\sigma$.

**Step 4:** Realizing layer-wise computation of $\Psi$.

The computations performed in a layer $\Psi^\ell$ of $\Psi$ are described in (8). Hence, for $1 \leq \ell < L$ the computation can be expressed as

$$\mathcal{R}_{\Psi^\ell}(y^{l-1}) = \sigma(A^l y^{l-1} + B^l) = \begin{pmatrix} \sigma(\sum_{i=1}^d A_{1,i}^l y_i^{l-1} + B_1^l) \\ \vdots \\ \sigma(\sum_{i=1}^d A_{d,i}^l y_i^{l-1} + B_d^l) \end{pmatrix} =: \begin{pmatrix} \sigma(f_1(y^{l-1})) \\ \vdots \\ \sigma(f_d(y^{l-1})) \end{pmatrix},$$

where $f_1^\ell, \ldots, f_d^\ell$ are affine linear functions with one-dimensional range on the same input domain $[a^{\ell-1}, b^{\ell-1}] \subset \mathbb{R}^d$, where $[a^0, b^0] = [a, b]$ and $[a^\ell, b^\ell]$ is the range of

$$(\sigma \circ f_1^{\ell-1}, \ldots, \sigma \circ f_d^{\ell-1})([a^{\ell-1}, b^{\ell-1}]^d).$$

Thus, via Step 3, we construct SNNs $\Phi_1^\ell, \ldots, \Phi_d^\ell$ that realize $\sigma \circ f_1^\ell, \ldots, \sigma \circ f_d^\ell$ on $[a^{\ell-1}, b^{\ell-1}]$. Note that by choosing appropriate parameters in the construction performed in Step 2 (as described below (26)), e.g., $\|A^l\|_\infty$ and $\|B^l\|_\infty$, we can employ the same input and output reference time for each $\Phi_1^\ell, \ldots, \Phi_d^\ell$. Consequently, we can parallelize $\Phi_1^\ell, \ldots, \Phi_d^\ell$ (see Lemma 2) and obtain networks $\Phi^\ell = P(\Phi_1^\ell, \ldots, \Phi_d^\ell)$ realizing $\mathcal{R}_{\Psi^\ell}$ on $[a^{\ell-1}, b^{\ell-1}]$. Finally, $\Psi^L$ can be directly realized via Step 2 by an SNN $\Phi^L$ (as in the last layer no activation function is applied and the output is one-dimensional). Although $\Phi^\ell$ already performs the desired task of realizing $\mathcal{R}_{\Psi^\ell}$ we can slightly simplify the network.

By construction in Step 3, each $\Phi_i^\ell$ contains two auxiliary neurons in the hidden layers. Since the input and output reference time is chosen consistently for $\Phi_1^\ell, \ldots, \Phi_d^\ell$, we observe that the auxiliary neurons in each $\Phi_i^\ell$ perform the same operations and have the same firing times. Therefore, without changing the realization of $\Phi^\ell$ we can remove the auxiliary neurons in $\Phi_2^\ell, \ldots, \Phi_d^\ell$ and introduce synapses from the auxiliary neurons in $\Phi_1^\ell$ accordingly. This is exemplarily demonstrated in Figure 2c for the case $d = 2$. After this modification, we observe that $L(\Phi^\ell) = L(\Phi_i^\ell) = 3$ and

$$
N(\Phi^\ell) = N(\Phi_1^\ell) + \sum_{i=2}^{d} \left( N(\Phi_i^\ell) - 2 - N_0(\Phi_i^\ell) \right) = dN(\Phi_1^\ell) - (d-1)(2 + N_0(\Phi_1^\ell))
$$
$$
= d(d+6) - 2(d-1) - (d-1)(d+1) = 4d + 3 \quad \text{for } 1 \le \ell < L,
$$

whereas $L(\Phi^L) = 1$ and $N(\Phi^L) = d + 2$.

**Step 5:** Realizing compositions of layer-wise computations of $\Psi$.
The last step is to compose the realizations $\mathcal{R}_{\Phi^1}, \ldots, \mathcal{R}_{\Phi^L}$ to obtain the realization

$$
\mathcal{R}_{\Phi^L} \circ \cdots \circ \mathcal{R}_{\Phi^1} = \mathcal{R}_{\Psi^L} \circ \cdots \circ \mathcal{R}_{\Psi^1} = \mathcal{R}_\Psi.
$$

As in Step 3, it suffices again to verify that the concatenation of the networks $\mathcal{R}_{\Phi^1}, \ldots, \mathcal{R}_{\Phi^L}$ is feasible. First, note that for $\ell = 1, \ldots, L$ the input domain of $\mathcal{R}_{\Phi^\ell}$ is given by $[a^{\ell-1}, b^{\ell-1}]$ so that, we can fix the suitable output reference time $T_{\text{out}}^{\Phi^\ell}$ based on the parameters of the network, the input domain $[a^{\ell-1}, b^{\ell-1}]$, and some input reference time $T_{\text{in}}^{\Phi^\ell} \in \mathbb{R}$. By construction in Steps 2 - 4 $T_{\text{in}}^{\Phi^\ell}$ can be chosen freely. Hence setting $T_{\text{in}}^{\Phi^{\ell+1}} = T_{\text{out}}^{\Phi^\ell}$ ensures that the reference times of the corresponding networks agree. It is left to align the input dimension of $\Phi^{\ell+1}$ and the output dimension of $\Phi^\ell$ for $\ell = 1, \ldots, L-1$. Due to the auxiliary neuron in the input layer of $\Phi^{\ell+1}$, we also need to introduce an auxiliary neuron in the output layer of $\Phi^\ell$ (see Figure 2d) with the required firing time $T_{\text{in}}^{\Phi^{\ell+1}} = T_{\text{out}}^{\Phi^\ell}$. Similarly, as in Step 3, it suffices to add a single synapse from the auxiliary neuron in the previous layer to obtain the desired firing time.

Thus, we conclude that $\Phi = \Phi^L \bullet \cdots \bullet \Phi^1$ realizes $\mathcal{R}_\Psi$ on $[a, b]$, as desired. The complexity of $\Phi$ in the number of layers and neurons is given by

$$
L(\Phi) = \sum_{\ell=1}^{L} L(\Phi^\ell) = 3L - 2 = 3L(\Psi) - 2
$$

and

$$
N(\Phi) = N(\Phi^1) + \sum_{\ell=2}^{L} \left( N(\Phi^\ell) - N_0(\Phi^\ell) \right) + (L-1)
$$
$$
= 4d + 3 + (L-2)(4d + 3 - (d+1)) + (d+2 - (d+1)) + (L-1)
$$
$$
= 3L(d+1) - (2d+1)
$$
$$
= N(\Psi) + L(2d+3) - (2d+2)
$$

$\square$

**Remark 7.** *Note that the delays play no significant role in the proof of the above theorem. Nevertheless, they can be employed to alter the timing of spikes, consequently impacting the firing time and the resulting output. However, the exact function of delays requires further investigation. The primary objective is to present a construction that proves the existence of a spiking network capable of accurately reproducing the output of any ReLU network.*

