# OpenReview forum: "Expressivity of Spiking Neural Networks through the Spike Response Model"
_NeurIPS.cc/2023/Workshop/UniReps — UniReps Oral_

### Official Review · Reviewer_xgjd · 2023-10-11

**Rating:** 8
**Confidence:** 4

**Review:**

This paper is well-motivated (and clear in its motivations) and effectively places itself in the context of the related literature.
It aims to address the lack of comprehensive theory in understanding the expressive power of SNNs, they do this by comparing with the expressive power of ANNs employing an piecewise linear activation function. Namely they characterize the number of 'linear regions' the input domain of the networks may be partitioned into (which is a proxy for expressivity). They show that an SNN has capacity to reproduce the output of any ReLU ANN (under specific conditions), and can realise certain continuous peicewise linear functions with fewer layers/computational units. The results are useful in the direction of designing architectures for implementation on neuromorphic hardware.

Some numerical simulations would help improve the paper

Clarity points:
* Add references to line 56, where other models of spike dynamics use differential equations
* In Eq (1) and all equations that follow, should the sum over edges not be restricted to the sum over edges connecting to neighbours of neuron v?

* $$\delta$$ confused me, how is it chosen? Its the length of a linear segment of the response function... which linear segment? Isn't the entire response function linear for t > 0?

* line 122 $$N$$ undefined, assuming number of parameters?
* a limitations section in conclusion would be good, but I understand page number is tight

Questions:
* Are there experiments/simulations that could be run to strenghthen the results (e.g. exponential scaling of the number of linear regions) in directly comparing expressivity
* Can the scaling result be extended to more than 1 layer SNNs?

---

### Official Review · Reviewer_PdjJ · 2023-10-23
**An interesting theory work on the expressiveness of SNN vs. ANN**

**Rating:** 7
**Confidence:** 1

**Review:**

Strengths:
1. Theoretical work studying the expressiveness of a family of SNNs in comparison with ANNs. Such works are very needed in the SNN community.
2. The theoretical results seem important to me, particularly Theorem 2.


Weakness:
1. There are no simulation results, not even on very simple datasets or simple function-fitting simulation results.
2. The spiking model assumption seems strong. I have concerns that is too far away from what SNN people usually use in experiments.

I do not have expertise in SNN theory nor did I carefully check the proofs.

---

### Official Review · Reviewer_F7SN · 2023-10-23
**well-written paper but lack of experimental demonstration**

**Rating:** 6
**Confidence:** 1

**Review:**

This paper looks into the relationship between Spiking Neural Networks (SNN) and Artificial Neural Networks (ANN) focusing on their expressive power on Continuous Piecewise Linear (CPWL) mappings, especially ReLU. The topic is engaging and the paper is well-written. The theorems, especially Theorem 4, which implies that SNN can have competitive expressive power even using fewer neurons, are interesting.

My concerns are:
1. The paper makes claims but doesn't show experiments to prove them. It would be stronger with some practical demonstrations.
2. The paper mentions the work's relevance to neuromorphic hardware design but doesn’t go into detail. A deeper discussion here would be useful.
3. The paper simplifies some things for easier analysis which may lessen the practical use of the findings.

---

### Decision · Program_Chairs · 2023-10-28

Accept (Oral)